# Causal Foundation Models with Continuous Treatments

**Christopher Stith** [* 1] **Medha Barath** [* 2] **Vahid Balazadeh** [2 3] **Jesse C. Cresswell** [1] **Rahul G. Krishnan** [2 3]

## Abstract

Estimating causal effects from observational data is a fundamental challenge in many disciplines. In this paper, we present the first *causal foundation model* for the continuous treatment setting. This setting is even more challenging as models need to represent effects across a continuum of treatment values. Our model meta-learns the ability to predict causal effects across a wide variety of unseen tasks without additional training or fine-tuning. We design a novel prior over data-generating processes with continuous treatments, then train a transformer to reconstruct individual treatment-response curves given only observational data, leveraging in-context learning to amortize expensive Bayesian posterior inference. Our model achieves state-of-the-art performance on individual treatment-response curve reconstruction tasks compared to causal models trained for each task.

## 1. Introduction

Causal inference is a central task for decision-making across many domains, including precision medicine, econometric policy-making, and algorithmic marketing. Estimating the effect of an intervention from observational data alone is complicated, as the presence of confounders can bias naive estimators of potential outcomes. The causal inference community has a rich library of estimators under the framework of ignorability, which assumes no unobserved confounding exists (Neal, 2020). Specifically, we assume:

1. **Unconfoundedness**. *Conditional on the covariates* $\mathbf{X}$, *the potential outcomes* $Y^t$ *are independent of the treatment assignment* $T$,

$$Y^t \perp T \mid \mathbf{X} \quad \forall t \in \mathcal{T}.$$

2. **Positivity/Overlap**. *There exists a constant* $c > 0$ *such that for all* $\mathbf{x} \in \mathcal{X}$ *and all* $t \in \mathcal{T}$,

$$p_{T|\mathbf{X}}(t \mid \mathbf{x}) \geq c.$$

*Equal contribution [1]Layer 6 AI, Toronto, Canada [2]University of Toronto [3]Vector Institute. Correspondence to: Chistopher Stith <christopher@layer6.ai>, Rahul G. Krishnan <rahulgk@cs.toronto.edu>.

*Proceedings of the $2^{nd}$ ICML Workshop on Foundation Models for Structured Data*, Seoul, South Korea. 2026. Copyright 2026 by the author(s).

However, the end-to-end implementation of existing estimators involves considerable time and effort: for any given task, a domain expert must inspect the data, propose an underlying mechanism to model it, choose an estimator that fits this mechanism, and only then train their model.

In addition to this approach, there has been recent work at the intersection of causal inference and meta-learning (Bynum et al., 2025). The goal here is to train a single model to perform a wide variety of causal inference tasks. A particularly promising framework has been Bayesian inference and in-context learning (ICL), as adopted by CausalPFN (Balazadeh et al., 2025), Do-PFN (Robertson et al., 2025), and CausalFM (Ma et al., 2026). Here, a model is trained over a diverse set of causal data-generating processes (DGPs) drawn from a prior $\pi(\psi)$ on possible DGPs $\psi$. At inference, observational data $\mathcal{D}_{\text{obs}}$ for a given *unseen* task is passed to the model as context, from which the model learns the posterior-predictive distribution for the causal estimand of interest, which is defined as

$$\pi^g(\cdot \mid \mathcal{D}_{\text{obs}}) := \left[ B \mapsto \int \mathbb{I}(g(\psi) \in B) \, \pi(\psi \mid \mathcal{D}_{\text{obs}}) \, \mathrm{d}\psi \right],$$
$$B \in \mathcal{B}, \quad (1)$$

where $\mathcal{B}$ is the Borel $\sigma$-algebra over $\mathbb{R}$. This approach amortizes the cost of posterior inference and turns the expensive manual implementation of causal inference into a completely data-driven process.

Almost all previous works in Bayesian causal inference, including the ICL approaches, focus on the binary treatment setting, where units can be split into control and treatment groups. However, numerous applications deal with the *continuous treatment* setting, in which interventional variables have a continuous range. In economics one is interested in the sensitivity of economic indicators to prices or rates, and in medical contexts one is interested not just in whether or not to treat a patient, but *how much* of a medication to give.

In the continuous treatment setting, we are specifically concerned with estimating the *individual treatment-response curve* (ITRC), which represents the causal effect of applying any of the continuous treatment levels on a unit with covariates $\mathbf{x}$. We define this curve using *conditional expected potential outcomes* (CEPOs). For a given covariate vector $\mathbf{x}$ and a treatment level $t$, the CEPO is defined as

$$\mu_t(\mathbf{x}) := \mathbb{E}[Y^t \mid \mathbf{X} = \mathbf{x}], \qquad \forall t \in \mathcal{T}. \quad (2)$$

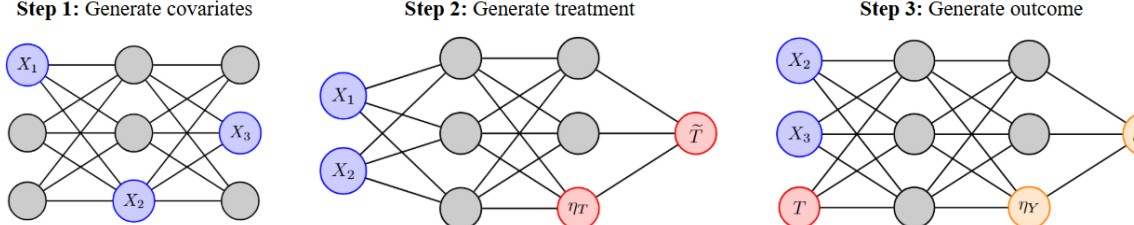

*Figure 1.* A schematic of our 3-MLP prior. In practice all MLPs drop edges with a certain probability.

The ITRC with covariates $\mathbf{x}$ is simply the function $\mathcal{T} \to \mathcal{Y}$ traced out by the CEPO across all treatment levels:

$$t \mapsto \mu_t(\mathbf{x}), \qquad \forall\, t \in \mathcal{T}. \tag{3}$$

The CEPO, and hence the ITRC, is *identifiable* when it can be written as a function of the observational distribution $P_{\text{obs}}$ (Neal, 2020). This is possible under the **Unconfoundedness** and **Positivity/Overlap** assumptions we make above.

Transitioning from binary to continuous treatments introduces significant theoretical and practical hurdles. Whereas in the binary case the goal is to estimate a single number per unit or per dataset (such as the Average Treatment Effect), continuous settings often require the estimation of a full *treatment-response curve*. This is a non-trivial task; many curves are consistent with the observed data, and without additional structural assumptions the treatment-response curve is not identified from observational data alone. Designing a prior that captures a suitably diverse set of continuous-valued DGPs is also highly non-trivial. The model must not only learn how to represent scalar effects, but also a wide spectrum of functions while remaining robust to the issues arising from the higher dimensionality of continuous treatment spaces. Finally, within a transformer-based architecture (Vaswani et al., 2017), representing continuous-valued treatments requires careful consideration. Standard tokenization or simple linear projections may fail to capture the high-frequency variations or local smoothness required for accurate causal discovery.

We introduce CCPFN (**C**ontinuous **C**ausal **P**rior-**F**itted **N**etwork), a causal foundation model that learns to reconstruct continuous ITRCs via ICL, and that can be applied off-the-shelf to any causal inference problem without fine-tuning. Central to this work is a novel prior over DGPs to generate synthetic covariates, treatments, and outcomes that directly encode identifiability. To summarize, our key contributions are:

- The first causal foundation model for continuous treatments, reaching top performance on ITRC reconstruction.
- A novel prior over DGPs which generates a rich set of synthetic causal data with continuous treatments.
- A synthetic causal scenario generation method for model validation in causal inference with continuous treatments.

## 2. Method

CCPFN was trained on a novel synthetic prior designed to address the challenges of the continuous treatment in causal inference. We design these priors so that the assumptions of unconfoundedness and positivity hold (see Section 1). A schematic of our prior is shown in Figure 1; it consists of three key components:

- **$\mathbf{MLP_X}$** generates the covariates $\mathbf{X}$, which can be divided into three disjoint subsets: the set $\mathbf{X}^T$ which are direct causes of $T$ only; the set $\mathbf{X}^Y$ which are direct causes of $Y$ only; and the set of confounders $\mathbf{X}^{\text{conf}}$ which are direct causes of both $T$ and $Y$. Noise is fed into each node during the forward pass, and random "tabular corruption" (binarization, quantization, zero-inflation) is applied both *during* and after the forward pass.
- **$\mathbf{MLP_T}$** generates the observed treatment $T$ from $\mathbf{X}^T \cup \mathbf{X}^{\text{conf}}$. The output layer is defined to be $\mathbb{E}[T \mid \mathbf{X}]$, and a random node $\eta_T$ is chosen to act as heteroscedastic mean-zero noise. The factual (observed) treatment is then set to be $T = \mathbb{E}[T \mid \mathbf{X}] + \eta_T(\mathbf{x})$.
- **$\mathbf{MLP_Y}$** generates outcomes (both factual and counterfactual), given treatment $t \in \mathcal{T}$ and covariates $\mathbf{X}^Y \cup \mathbf{X}^{\text{conf}}$. As in $\mathbf{MLP_T}$, the output layer is the CEPO $\mu_t(\mathbf{x})$, and a random node $\eta_Y$ is chosen to be heteroscedastic noise. We then set the factual outcome to be $Y = \mu_T(\mathbf{x}) + \eta_Y(\mathbf{x})$.

The structural causal model (SCM) built up from these three MLPs produces samples $\left\{\left(\mathbf{x}_n, t_n, y_n, t'_n, \mu_{t'_n}(\mathbf{x}_n)\right)\right\}_{n=1}^{N}$, where $(t_n, y_n)$ are the factual treatment and outcome, respectively, $t'_n$ is a counterfactual treatment, and $\mu_{t'_n}(\mathbf{x}_n)$ is the corresponding CEPO. This constructs a prior over possible scenarios which satisfy the *backdoor assumption* of causal inference.

We use a similar architecture as in CausalPFN (Balazadeh et al., 2025), with a PFN-style transformer encoder that leverages ICL to learn the parameterized CEPO-PPD $q_\theta$. The central new architectural piece of CCPFN is a separate encoder for treatments to ensure that the signal from the treatment variable is not lost in high-dimensional settings. Input tokens $t$ are passed directly through a nonlinear $T$-encoder, and in addition are separately appended to $\mathbf{x}$ and passed through a linear encoder (reminiscent of S-Learners (Künzel et al., 2019)); see Figure 2. Moreover, as we work

*Table 1.* Comparative evaluation of Mean MISE across benchmark test datasets. Values represent mean MISE $\pm$ standard deviation as computed with 5-fold cross-validation. First place is **bold**, second place is underlined. Dashes — indicate no meaningful results were obtained. When evaluating TabPFN, due to memory constraints and to match the dimensionality reduction used in CCPFN, we applied PCA to reduce the dimension to 100. DRNet, VCNet, and EBCT did not produce meaningful MISE results and hence are omitted.

| Method | Mean MISE ($\downarrow$ better) | | | | | | |
|---|---|---|---|---|---|---|---|
| | MVICU ($\times 10^3$) | Debt ($\times 10^{-2}$) | Warfarin | TCGA | News | NewsHet ($\times 10^{-2}$) | Avg. Rank |
| **CCPFN (Ours)** | $\underline{1.45 \pm 2.4}$ | $\mathbf{2.26 \pm .19}$ | $\mathbf{40.4 \pm 5.7}$ | $8.63 \pm 1.8$ | $1.58 \pm .11$ | $5.76 \pm .39$ | **2.8** |
| S-Learner (TabDPT) | $1.53 \pm 2.5$ | $\underline{2.41 \pm 0.22}$ | $\underline{47.4 \pm 8.1}$ | $8.47 \pm 1.3$ | $\underline{1.55 \pm .05}$ | $6.28 \pm .16$ | $\underline{3.3}$ |
| S-Learner (TabPFN) | $1.59 \pm 2.4$ | $5.80 \pm 0.51$ | $159 \pm 37$ | $5.61 \pm .90$ | $\mathbf{1.49 \pm .06}$ | $6.33 \pm .11$ | $4.3$ |
| S-Learner (TabICL) | $\mathbf{1.49 \pm 2.5}$ | $3.70 \pm 0.59$ | $54.8 \pm 11$ | $\underline{4.01 \pm .94}$ | $1.68 \pm .05$ | $5.38 \pm .14$ | $3.5$ |
| ADMIT | $\mathbf{.250 \pm .064}$ | $2.64 \pm .33$ | $87.9 \pm 60$ | $6.21 \pm 1.7$ | $1.67 \pm .10$ | $8.12 \pm .32$ | $4.3$ |
| SCIGAN | $2.61 \pm 2.2$ | $4.44 \pm .07$ | $711 \pm 44$ | $\mathbf{3.75 \pm .91}$ | $1.88 \pm .09$ | $\underline{4.72 \pm .41}$ | $5$ |
| GPS | $3.09 \pm 3.0$ | $16.8 \pm .32$ | $418 \pm 49$ | — | $1.70 \pm .09$ | $6.36 \pm .13$ | $6.8$ |
| CausalForest | $70.7 \pm 0.0$ | $28.9 \pm 0.0$ | $894 \pm 24$ | $8.71 \pm 7.9$ | $1.78 \pm .57$ | $\mathbf{4.50 \pm .05}$ | $6.7$ |
| NonparamDML | $96.0 \pm 0.0$ | $39.4 \pm .57$ | $1230 \pm 49$ | $493 \pm 4.6$ | $1.63 \pm .39$ | $7.19 \pm .05$ | $7.8$ |

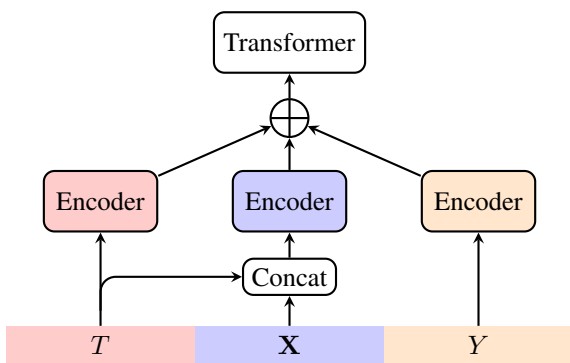

*Figure 2.* Illustration of the tri-encoder scheme used by CCPFN. Treatments $T$ are additionally routed through a separate encoder to boost treatment signal.

in the continuous treatment setting, we $z$-standardize all outcomes together (rather than splitting into control/treatment groups) as well as all treatment values. Given a query unit $\mathbf{x}$, a grid of treatment values can be passed to CCPFN to *immediately* estimate the full range of the ITRC $t \mapsto \mu_t(\mathbf{x})$.

**Training.** At each step of training, a DGP $\psi \sim \pi$ is sampled to yield a SCM which is generated by the three MLPs described above. We generate a dataset $\mathcal{D}$ of both factual and counterfactual scenarios. Counterfactual treatments are sampled uniformly on $[0, 1]$, and the model is asked to predict the CEPO for each $(\mathbf{x}_i, t')$ query pair. We use a *causal data-prior loss*. For any $t \in \mathcal{T}$, this loss is defined as

$$\mathcal{L}_t(\theta) := \mathbb{E}_{\psi \sim \pi} \left[ -\log q_\theta(\mu_t(\mathbf{x}\,;\,\psi) \mid \mathbf{x}, t, \mathcal{D}_{\text{obs}}) \right] \quad (4)$$

Further details can be found in Appendix D.1. In practice during training, we utilize a discretized version of (4). This ensures that the loss in (4) is minimized over the whole treatment range $\mathcal{T} = [0, 1]$. At training time, the dataset is randomly shuffled and a context/query split position $M$ is chosen. The observational dataset $\mathcal{D}_{\text{obs}} = \{(\mathbf{x}_i, t_i, y_i)\}_{i=1}^M$ is passed as context, and the counterfactuals $\{(\mathbf{x}_i, t_i')\}_{i=M+1}^N$ are passed as the query.

## 3. Experimental Results

We evaluate CCPFN against several baseline causal inference models that are trained on each task (Appendix A). In addition, we test top tabular foundation models implemented as S-Learners that are not trained per-task, using TabDPT (Ma et al., 2025b), TabPFN-2.6 (Grinsztajn et al., 2025), and TabICLv2 (Qu et al., 2026).

Evaluation is done on a suite of semi-synthetic benchmarks from a wide variety of domains including medicine, finance, and user research. We use the datasets MVICU, Debt, News, NewsHet, TCGA, and Warfarin. See Appendix B.1 for complete details.

To address the lack of benchmarks in the continuous treatment regime, we created a method to generate (semi)-synthetic causal scenarios. For each dataset, the method generates a causal inference scenario from descriptions of the data, a propensity model based on the scenario, and a treatment-response function (see Appendix B.2).

We evaluate our model's ability to perform two key tasks: 1) Reconstruct individual treatment-response curves, and 2) Prescribe the optimal treatment level/dosage for each individual. To evaluate performance on the first task, we use the mean integrated square error (MISE) (Schwab et al., 2020) between the true treatment-response curve $\mu_t$ and the model's predicted $\hat{\mu}_t$, which is reported as the mean of the CEPO-PPD $q_\theta(t, \mathbf{x})$. This is calculated over all $N$ individuals in a given dataset and over the whole treatment range $[a, b]$:

$$\text{MISE} = \frac{1}{N} \sum_{n=1}^N \frac{1}{b - a} \int_a^b \left| \hat{\mu}_t(\mathbf{x}_n) - \mu_t(\mathbf{x}_n) \right|^2 dt, \quad (5)$$

where $\mathbf{x}_n$ is the covariate vector of the $n$th individual. To evaluate performance on the second task, we use the mean dosage policy error (DPE) (Schwab et al., 2020). This is the

*Table 2.* Comparative evaluation of Mean DPE across benchmark test datasets. Values represent mean DPE $\pm$ standard deviation as computed with 5-fold cross-validation. First place is **bold**, second place is underlined. When evaluating TabPFN, due to memory constraints and to match the dimensionality reduction used in CCPFN, we applied PCA to reduce the dimension to 100.

| Method | Mean DPE ($\downarrow$ better) | | | | | |
| --- | --- | --- | --- | --- | --- | --- |
| | Debt ($\times 10^{-3}$) | Warfarin | TCGA | News | NewsHet ($\times 10^{-3}$) | Avg. Rank |
| **CCPFN (Ours)** | $0.29 \pm 0.22$ | $2.91 \pm 1.67$ | $38.90 \pm 8.10$ | $3.71 \pm 0.60$ | $1.64 \pm 0.57$ | 6 |
| S-Learner (TabDPT) | $8.24 \pm 3.15$ | $2.45 \pm 1.01$ | $35.50 \pm 8.19$ | $3.76 \pm 0.59$ | $0.74 \pm 0.32$ | 5 |
| S-Learner (TabPFN) | $72.50 \pm 13.4$ | $1.92 \pm 0.82$ | $35.48 \pm 8.29$ | $\mathbf{2.69 \pm 0.30}$ | $1.14 \pm 0.48$ | 4.8 |
| S-Learner (TabICL) | $14.37 \pm 6.19$ | $0.31 \pm 0.21$ | $35.8 \pm 6.7$ | $3.78 \pm .41$ | $\mathbf{.292 \pm .064}$ | 4.6 |
| ADMIT | $0.26 \pm 0.50$ | $\mathbf{0.10 \pm 0.08}$ | $24.6 \pm 5.4$ | $3.65 \pm 0.67$ | $0.63 \pm 0.04$ | **2.2** |
| SCIGAN | $\mathbf{0.00 \pm 0.0}$ | $150 \pm 150$ | $38.6 \pm 89$ | $3.88 \pm 0.54$ | $6.91 \pm 0.57$ | 7.6 |
| GPS | $678 \pm 180$ | $80.9 \pm 2.6$ | — | $3.46 \pm 0.38$ | $1.04 \pm 1.21$ | 6.3 |
| CausalForest | $\mathbf{0.00 \pm 0.0}$ | $2010 \pm 24$ | $33.9 \pm 7.9$ | $4.37 \pm .57$ | $6.84 \pm .47$ | 7.2 |
| NonparamDML | $28.2 \pm 5.7$ | $2000 \pm 49$ | $\mathbf{23.6 \pm 4.6}$ | $3.81 \pm .39$ | $6.84 \pm .47$ | 7 |
| DRNet | $64.7 \pm 70$ | $2.19 \pm 1.5$ | $29.5 \pm 6.6$ | $6.42 \pm 2.0$ | $5.35 \pm 4.60$ | 6.8 |
| VCNet | $207 \pm 140$ | $1800 \pm 1000$ | $31.6 \pm 12$ | $5.00 \pm 1.13$ | $6.84 \pm 0.47$ | 8.6 |
| EBCT | $327 \pm 240$ | $1400 \pm 1300$ | $45.0 \pm 12$ | $6.42 \pm 3.2$ | $1.91 \pm 2.78$ | 9.8 |

error between the true CEPO at the true optimal dosage/treatment level $t^*$ and the true CEPO at the model's estimated optimal dosage $\hat{t}^*$, averaged over the whole dataset:

$$\text{DPE} = \frac{1}{N} \sum_{n=1}^{N} \left( \mu_{t^*(\mathbf{x}_n)}(\mathbf{x}_n) - \mu_{\hat{t}^*(\mathbf{x}_n)}(\mathbf{x}_n) \right)^2, \quad (6)$$

where $t^*(\mathbf{x})$ is the optimal dosage for an individual with covariates $\mathbf{x}$. We consider in this paper optima that are either global minima or maxima, depending on the scenario.

CCPFN demonstrates superior performance on the ITRC prediction task across multiple benchmark datasets (Table 1). When evaluated by average rank relative to the Mean Integrated Squared Error (MISE), CCPFN comes in first place overall. Our model remains competitive on the dosage policy task (Table 2). Notably, the performance of neural-network and statistical baselines varies significantly across datasets, suggesting a reliance on extensive hyperparameter tuning. On the other hand, CCPFN yields consistent results without per dataset weight updates.

To qualitatively assess the model's ability to capture complex treatment-response dynamics, we visualize its predicted curves against the ground truth Individual Treatment-Response Curve (ITRC). Figure 3 illustrates the predictions of the five top-performing methods on the Warfarin dataset (Kallus & Zhou, 2018). While all methods capture the general V-shape, CCPFN (shown in blue) tracks the ground truth more closely, particularly demonstrating superior accuracy near the endpoints of the dosage range.

We perform ablations on particular choices made in the design of our prior. We find a significant performance gain from including in-pass tabular "corruption" (e.g. binarizing, quantizing, or zero-inflating randomly-selected nodes) as opposed to solely post-hoc (Table 3). See Appendix E for further details.

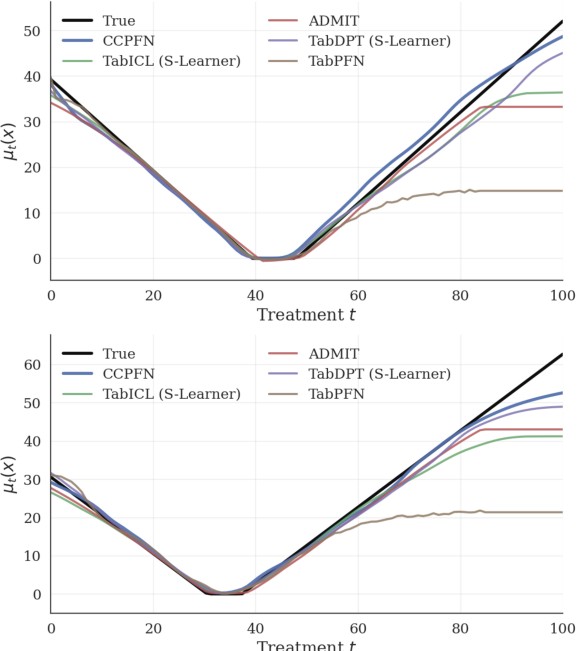

*Figure 3.* Selected Individuals' Predicted ITRCs on Warfarin.

## 4. Conclusions

We introduce CCPFN, a causal foundation model for the continuous treatment setting. It demonstrates superior ability to reconstruct continuous individual treatment-response curves without any further fine-tuning on unseen datasets. A central contribution is our novel 3-MLP prior which naturally generates causal DGPs satisfying unconfoundedness and positivity, two properties that ensure the identifiability of treatment-response curves from observational data. Our work extends the capabilities of ICL-based Bayesian causal inference, such as CausalPFN, Do-PFN, and CausalFM.

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

# A. Related Work

While ours is the first model to leverage in-context learning for the continuous treatment regime in causal inference, numerous techniques have been developed over the past decade to address this regime due to its importance across a variety of domains. DRNet (Schwab et al., 2020) addresses the challenge of multi-armed treatments, each with associated treatment parameters. For each treatment, DRNet breaks the treatment range into $E \in \mathbb{N}$ equally-sized subintervals and assigns a head to each. This ensures the effect of the treatment parameter persists throughout the network layers, but it can lead to discontinuities in the learned treatment-response curves. SCIGAN (Bica et al., 2020) leverages generative adversarial networks (Goodfellow et al., 2014) to generate and learn counterfactual representations. VCNet (Nie et al., 2021) further addresses the continuous treatment space by training a varying coefficient network, where neural network parameters $\theta = \theta(t)$ depend on the treatment value $t$. ADMIT (Wang et al., 2022) extends the theoretical framework by bounding the counterfactual loss of estimating treatment-response curves and proposes an algorithm that makes counterfactual estimations.

Meanwhile, several groups have approached the binary treatment setting for causal inference via causal foundation modeling; see (Balazadeh et al., 2025; Robertson et al., 2025; Ma et al., 2026).

# B. Further Details About Benchmarks and Baselines

## B.1. Benchmark Details

- **MVICU.** A semi-synthetic medical scenario adapted from (Schwab et al., 2020) and post-processed. It models the effect of different configurations of mechanical ventilation in the intensive care unit on patients. For treatment and outcome functions, we used the same ones as in News (Wang et al., 2022). It contains 13 covariates and 4,963 rows. (PhysioNet Credentialed Health Data License 1.5.0)
- **Debt.** A fully synthetic financial scenario concerning the impact of different levels of debt write-downs on total debt repaid, adapted from (Moral Hernández et al., 2025). The generated dataset has 10 covariates and 10,000 rows. (MIT license)
- **News.** A semi-synthetic scenario representing reader interaction with news articles from (Schwab et al., 2020; Wang et al., 2022). It has 2,870 covariates. Using the original dataset downloaded from (Schwab et al., 2020), we use the process outlined in (Wang et al., 2022) to process the data. Specifically, we randomly subsampled 10,000 rows. From this, we further subsampled 7,881 by dropping rows whose optimal CEPO (as determined by the oracle) was greater than 10. (MIT license)
- **NewsHet.** A semi-synthetic scenario adapted from News, with treatment-response functions modified to introduce heterogeneity of optimal dosage levels. As implemented in (Wang et al., 2022), News suffers from the fact that all optima occur at the endpoints 0 or 1, making the optimal dosage policy prediction a degenerate task. (MIT license)
- **TCGA.** A semi-synthetic scenario representing the effect of medication dosage and risk of cancer recurrence from (Schwab et al., 2020; Wang et al., 2022), where we use the DGP outlined by (Wang et al., 2022) to generate ground-truth causal effects. We additionally removed rows whose optimal CEPO was greater than 50, leaving 4,428 rows. (MIT license)
- **Warfarin.** A semi-synthetic medical scenario concerning warfarin dosing, adapted from (International Warfarin Pharmacogenetics Consortium, 2009; Whirl-Carrillo et al., 2021; Kallus & Zhou, 2018). The dataset specifically calculates the loss between the actual dosage received by an individual and the optimal warfarin dosage calculated by the IWPC pharmacogenetic algorithm (International Warfarin Pharmacogenetics Consortium, 2009) and uses this as the outcome. The generated dataset has 19 covariates and 4,490 rows. (CC BY-SA 4.0)

## B.2. Synthetic and Semi-Synthetic Data Scenarios

While in the binary treatment regime public benchmarks have become more available recently, there is a serious lack of a broad suite of benchmarks in the continuous treatment regime. Part of the difficulty lies in having to store a potentially infinite number of counterfactuals per individual. To address these issues, we created a `Scenario` class to represent a causal inference scenario/context (for example, the administration of a particular drug to a particular set of patients), complete with a propensity model and a treatment-response function.

Each subclass of `Scenario` at minimum implements the following three methods:

- `load_covariates`: Generates the base covariates $\mathcal{D}_{\text{cov}} = \{\mathbf{x}_n\}_{n=1}^{N}$. For semi-synthetic data, these are loaded from real data; for fully synthetic data, these are generated synthetically.

- `treatment`: Generates the treatment administered to each individual.
- `dose_response`: The noise-free treatment-response function $(\mathbf{x}, t) \mapsto \mu_t(\mathbf{x})$ for this `Scenario`.

### B.3. Hyperparameter Optimization

**EconML Baselines** We tune results from the EconML Baselines (CausalForestDML and NonParamDML) using the FLAML (AutoML) library. Hyperparameter tuning is performed on the treatment model and the outcome model. We use K-fold cross-validation with K = 5, early stopping, and a time budget of 900 seconds. The following base estimators are used: `"lgbm"`, `"xgboost"`, `"xgb_limitdepth"`, `"rf"`, `"kneighbour"`, `"extra_tree"`.

**Neural Network Methods** For the neural network baselines, we conduct a grid search over learning rates $\in \{0.005, 0.001, 0.0005, 0.0003, 0.0001\}$ and batch sizes $\in \{128, 256, 512, 1024\}$. For each dataset, we sample 80% of the data and perform 5-fold cross-validation on this sample to identify the hyperparameter configuration yielding the lowest average training loss for each baseline. We use the training loss because in practice, one *never* has access to the synthetic validation metrics used. Once the optimal hyperparameters are identified, we perform 5-fold cross-validation on the entire training dataset to elicit final performance.

### B.4. Evaluation Protocol

The fully optimized versions of all considered models (CCPFN, TFMs, neural networks, EconML, and statistical baselines) are evaluated on the benchmark datasets using 5-fold cross-validation. For the neural network, statistical, and EconML baselines, the training splits are used to explicitly fit the models to each benchmark scenario. Conversely, for CCPFN and TFMs, these training splits are provided solely as in-context examples, leaving the models' underlying weights frozen. The final reported MISE and DPE metrics for each method represent the average performance across the five validation folds.

## C. Details on Priors

### C.1. Further Details on 3-MLP Prior

During training, the number of samples $N$ is fixed to be 2048. The maximum number of covariates is 98. The number of layers $L_{\mathbf{X}}, L_T, L_Y$ are all sampled from independent truncated normal distributions with mean and variance $\alpha > 0$, where $\alpha \sim \text{LogUniform}(A, B)$. In our experiments, we set $A = 1, B = 10$ for $\mathbf{X}, T$, and $Y$. The number of hidden units $H_{\mathbf{X}}, H_T, H_Y$ are also sampled from the same class of distributions, with $A = 10, B = 100$ for $\mathbf{X}, T$, and $Y$. The densities $d_{\mathbf{X}}, d_T, d_Y$ were sampled independently from $\mathcal{U}(0.1, 1)$. To ensure that the signal from $T$ is not lost when dropping edges, we protect all outgoing edges from $T$ in the input layer of $\mathbf{MLP}_Y$. In the truncated normal distributions, we truncate to $[3, \infty)$ for layer distributions and $[4, \infty)$ for hidden size distributions. Integer values are then obtained by rounding.

Below we include more detailed descriptions of each of the three MLPs in our prior.

$\mathbf{MLP}_{\mathbf{X}}$. In addition to $L_{\mathbf{X}}, H_{\mathbf{X}}$, and $d_{\mathbf{X}}$, a noise scale $s > 0$ and a natural number $K > 0$ (representing the number of covariates) are chosen. For each $l \in [L_{\mathbf{X}}]$, noise values $\epsilon^{(l)} \in \mathbb{R}^N \times \mathbb{R}^{H_{\mathbf{x}}}$ for each $n \in [N]$ are generated by randomly-chosen distributions in $\{\mathcal{N}(0, 1), \text{Laplace}(0, 1), t_3\}$. The input $\epsilon^{(0)}$ is propagated through the MLP by randomly initialized weights:

$$z^{(l)} = \sigma^{(l)}(W^{(l)}z^{(l-1)}) + \epsilon^{(l)}, \quad z^{(0)} = \epsilon^{(0)}, \tag{7}$$

where each $\sigma^{(l)}$ represents a vector of random activation functions. The MLP weights come from the normal distribution $\mathcal{N}(0, \sigma_w^2)$, where $\sigma_w = \left(\frac{2}{\max(H*p_d, 1.0)}\right)^{\frac{1}{2}}$; $H$ is the hidden width of the MLP while $p_d$ represents the probability of a given weight being kept by a random sparsity mask. We then choose $K$ nodes among the $z^{(l)}$ to be covariates; this produces covariates differing in complexity and marginal distribution. In addition, to simulate realistic tabular data, we apply random "tabular corruption". A random subset of covariate nodes is either binarized, quantized, or zero-inflated (across the $N$ samples). Contrary to past works (Hollmann et al., 2023; Qu et al., 2026), which only apply this transformation *after* the forward pass, we transform roughly 35% of covariate nodes *during* the forward pass. This allows the SCM to see and utilize the realistic tabular features during generation, adding realistic tabular diversity to the DGP itself, rather than solely post-hoc. Another roughly 65% of the remaining covariates are further corrupted after the forward pass is completed. The superiority of this method as opposed to solely post-hoc tabular corruption, is demonstrated in our ablations (see Table 3). This does not affect the theoretical validity of the SCM, as all tabular corruption occurs within $\mathbf{MLP}_{\mathbf{X}}$ alone; it merely enhances the diversity of the prior.

**MLP$_T$.** The input to **MLP**$_T$ is $\mathbf{X}^T \cup \mathbf{X}^{\text{conf}}$, and the output is a scalar $\tilde{T}$ which is the expected treatment value given $\mathbf{X}^T \cup \mathbf{X}^{\text{conf}}$. A random node $\eta_T$ in the hidden layers is chosen to act as a heteroscedastic noise scale parameter. To produce the final outcome $T$ we compute

$$T = \tilde{T} + \sigma(\tilde{T}) \cdot \eta_T \cdot \epsilon, \qquad \epsilon \sim \mathcal{N}(0, 1), \tag{8}$$

where $\sigma(\tilde{T})$ is the empirical standard deviation of the $\tilde{T}$ over the $N$ samples. Rather than designating the output node as $T$ itself, this method practically ensures that positivity $p(T \mid \mathbf{X} = \mathbf{x}) > 0$ holds, as $T$ is centered at $\tilde{T}$ with strictly positive variance. Finally, the treatment $T$ is min-max scaled to lie in $[0, 1]$.

**MLP$_Y$.** The input to **MLP**$_Y$ is $\mathbf{X}^Y \cup \mathbf{X}^{\text{conf}}$ as well as a treatment value $t \in [0, 1]$. This MLP thus generates both factual and counterfactual outcomes. Similar to **MLP**$_T$, the output is a scalar which is the CEPO $\mu_t(\mathbf{X})$, while a random node $\eta_Y$ in the hidden layers is chosen as noise scale parameter. During the factual forward pass, the output $T$ of **MLP**$_T$ is passed as input, and the factual outcome $Y$ is generated by

$$Y = \mu_T(\mathbf{X}) + \sigma(\mu) \cdot \eta_Y \cdot \epsilon, \qquad \epsilon \sim \mathcal{N}(0, 1), \tag{9}$$

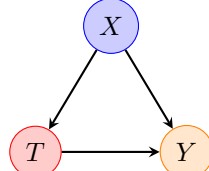

*Figure 4.* Causal graph associated with the backdoor setting.

where $\sigma(\mu)$ is the empirical standard deviation of the CEPOs over the $N$ samples. To generate CEPO for counterfactual treatment values $t \neq T$, one passes $t$ as input and selects the output node $\mu_t(\mathbf{X})$ without adding the noise $\eta_Y$.

## C.2. Alternative Priors

We compare two alternative prior distributions over data-generating processes.

**Bernstein polynomial prior.** We sample a table of $N$ rows and partition its columns into covariates $\mathbf{X}$ and a noise column $\epsilon$. Covariates are standardised, and treatments are drawn from a sigmoid-normal distribution,

$$t \mid \mathbf{x} \sim \sigma\Big(\mathcal{N}(\mu_{t\mid\mathbf{x}}, \sigma^2_{t\mid\mathbf{x}})\Big), \tag{10}$$

where $\mu_{t\mid\mathbf{x}}$ and $\sigma_{t\mid\mathbf{x}}$ are the outputs of a conditional MLP applied row-wise to $\mathbf{x}$. Overlap is controlled by scaling $\sigma_{t\mid\mathbf{x}}$ by an overlap parameter $\alpha \in (0, 1]$. The conditional expected potential outcome (CEPO) $\mu_t(\mathbf{x})$ is defined via a $K$-degree Bernstein polynomial, whose coefficients are formed as a convex combination of individual-specific coefficients $\mathbf{c}(\mathbf{x})$ (produced by a second conditional MLP) and a shared global coefficient vector $\mathbf{c}_0 \sim \mathcal{N}(\mathbf{0}, \mathbf{I})$:

$$\mathbf{c} = \lambda \mathbf{c}(\mathbf{x}) + (1 - \lambda) \mathbf{c}_0, \tag{11}$$

where $\lambda \in [0, 1]$ governs the degree of treatment effect heterogeneity. The observed outcome is then $y = \mu_t(\mathbf{x}) + \epsilon$, where $\epsilon$ is a scaled, centred noise term derived from the reserved noise column.

**Value-based prior.** We again sample a table of $N$ rows and select a subset of columns as covariates $\mathbf{X}$. Rather than parameterising the treatment-response curve analytically, we directly read off potential outcomes at $n$ randomly sampled (and sorted) treatment support points $\{t_1, \ldots, t_n\} \subset [0, 1]$. For each support point $t_k$, we reserve one table column as the CEPO $\mu_{t_k}(\mathbf{x})$ and a second column as an individual-level noise term $\eta_k(\mathbf{x})$; noise columns are scaled so that their contribution is a fixed fraction of the corresponding signal variance. Observed outcomes at any treatment value $t$ are obtained by linearly interpolating both $\mu_{t_k}(\mathbf{x})$ and $\eta_k(\mathbf{x})$ across the two nearest support points and summing the results. Treatments are sampled using the same sigmoid-normal mechanism as in the Bernstein prior.

# D. Model Training Details

## D.1. Model Architecture and Hyperparameters

We use a modified version of the TabDPT architecture (Ma et al., 2025b) with a nonlinear $T$-encoder (see Figure 2). We initialize our weights to TabDPT's trained weights on layers which support them. When the number of covariates $K$ exceeds 100, we apply truncated SVD to the covariates before passing them to the model. During training, we used a batch size of 32, 8 gradient accumulation steps, and 128 model updates per epoch. Our final model was fine-tuned for 15 epochs; this number was selected by early stopping based on the validation dataset results.

As mentioned in Section 2, instead of the exact causal data-prior loss (4), we use the *histogram loss* (12). We employ regression-as-classification and quantize the outcome variable into $L = 1024$ bins from $[-10, 10]$ (after $z$-standardization). The model then outputs $q_\theta$ as a discrete histogram distribution on each of these $L$ bins. To approximate the true CEPO-PPD in a tractable manner we assume it is given by $\mathcal{N}(\mu_t(\mathbf{x}), \sigma^2)$ for $\sigma \ll 1$. The *histogram loss* which we train with is then defined as

$$\mathcal{L}_{\text{HL}}(\theta \mid \mathbf{x}, t) = -\sum_{l=1}^{L} \mathcal{N}(\mu_t(\mathbf{x}), \sigma^2)[l] \log q_\theta(l \mid \mathbf{x}, t, \mathcal{D}_{\text{obs}}), \tag{12}$$

---

**Config 1** Configuration for CCPFN.

---

```
n_out: 10
nbins: 1024
nhead: 6
nhid: 768
ninp: 384
nlayers: 20
num_covariates: 100
```

---

The final model architecture has 19,140,352 parameters. Of these, 297,600 are the nonlinear $T$-encoder parameters, compared to 39,168 for the linear $\mathbf{X}$-and-$T$-encoder and 768 for the $Y$-encoder.

All training runs were performed on single NVIDIA A6000 GPUs.

### D.2. Validation Dataset Construction

The datasets we used for model validation and hyperparameter selection were a mix of synthetic and semi-synthetic. We used the fully synthetic dataset constructed in (Wang et al., 2022), as well as a simple linear dataset with a linear treatment-response function. We chose the best version of our model based on averaged IQR-normalised MISE performance across all validation datasets.

For semi-synthetic data, we used covariates from several well-known causal inference datasets: ACIC2016 (Apache license) (Dorie et al., 2019), ACIC2018 (Apache license) (Karavani et al., 2018), Criteo (MIT license) (Eustache Diemert, Artem Betlei et al., 2018), Hillstrom (MIT license) (Hillstrom, 2008), Lalonde (MIT License) (Dehejia & Wahba, 1999; Huntington-Klein, 2024), Lenta (MIT license) (Lenta & Microsoft, 2020; Shevchenko & contributors, 2020), Twins (MIT license) (Neal et al., 2021), and X5 (MIT license) (X5 Retail Group & ODS.ai, 2019; Shevchenko & contributors, 2020). We tasked an LLM agent with generating a plausible scenario for each set of covariates using the prompts included below.

---

**System Prompt for Generating Synthetic Validation Data #1**

```
# Semi-synthetic data generation instructions

(Note: run python scripts in the `conda tracee` environment.)

## Background
You are working on a project in causal inference. The goal is to train a model to
    perform causal inference in the *continuous treatment* setting.

## Your Task
Your task is to create semi-synthetic data consisting of real-world covariates `X` (
    real data) and synthetic treatment and outcome variables by creating synthetic
    data-generating processes (DGPs). Adhere to the following instructions:

1. Ask the user which `csv` file to use as the base covariates `X`. It is possible
    that there is no local csv, and the covariates will have to be downloaded in the
     script itself (e.g. using sklearn.datasets). You can download and view the
    covariates now, so that you have intuition for the context.
```

---

```
  2. Ask the user for covariate context, i.e. what do the base covariates `X`
     represent in this dataset?
  3. Ask the user for treatment and outcome context, i.e. what scenario the user has
     in mind for the treatment and outcomes.
  4. Based on the information provided in steps 1 – 3, devise a *realistic* DGP to
     simulate treatment assignment and outcomes. Remember, the treatment variable
     should be continuous, *not* binary. This DGP should satisfy the following
     requirements:
   1. There should be a high degree of confounding: at least 50% of the covariates
      should be causes of both the treatment and the outcome.
   2. You should generate a *dose-response function* f(X, t) that maps an individual
      with covariates X and hypothetical treatment t to the *conditional expected
      potential outcome (CEPO)*. This should be a suitably complex and realistic
      function which can be implemented in simple Python code.
   3. You should generate a *treatment assignment function* T(X) that maps an
      individual with covariates X to the *observed* treatment T(X). This should be a
      suitably complex and realistic function which can be implemented in simple
      Python code.
   4. In order to ensure that there is a high degree of confounding, the functions f
      and T should both depend on some subset of covariates comprising at least half
      of the total number of covariate features.
  5. Once you have constructed this DGP, generate a *Python script* that outputs a csv
     file as follows:
   1. Ask the user for the desired name of the Python script.
   2. The Python script should include code for the dose-response function f(X, t) and
      the treatment assignment function T(X).
   3. The Python script should output a single csv file with columns named x_0 through
      x_n (where n is the number of covariate features), t, y, t_test, cepo_test.
      The data should be filled as follows:
    1. The values of columns x_0 through x_n should be the values of the original
       base covariates csv.
    2. The value of t should be the value of T(X) for X the corresponding covariate
       value.
    3. The value of y should be f(X, t) for X the corresponding covariate value and t
       = T(X), *plus Gaussian noise* which is iid for each row.
    4. The value of t_test should be randomly sampled from [t_min, t_max].
    5. The value of cepo_test should be f(X, t_test).
    6. All data should be numerical (e.g. string-based categorical variables should
       be encoded as integers).
  4. Save the python script in tracee/inference/benchmarks/data_generation_scripts.

When you are ready to proceed with this task, begin at step 1 above.
```

## System Prompt for Generating Synthetic Validation Data #2

```
#Semi-synthetic data generation instructions

##Background

You are working on a project in causal inference. The goal is to train a model to
    perform causal inference in the continuous treatment setting. ##Your Task Your
    task is to create semi-synthetic data consisting of real-world covariates X (
    real data) and synthetic treatment and outcome variables by creating synthetic
    data-generating processes (DGPs). The dataset we are working with is the Lalonde
     dataset. You can decide the best way to access this dataset (realcause, jobs
    etc.) Only use the base covariates. Propose a potential CONTINUOUS-VALUED
    treatment (T) and CONTINUOUS-VALUED outcome (Y). You should then realise a DGP
    to generate this treatment and outcome. Some things to keep in mind are:

We are working in the backdoor causal graph scenario, where X -> T, T -> Y, and X->
```

```
    Y

There should be a high degree of confounding - at least 50% of the covariates should
    affect outcome Y You should generate a dose-response function f(X, t) that maps
    an individual with covariates X and hypothetical treatment t to the conditional
    expected potential outcome (CEPO). This should be a suitably complex and
    realistic function which can be implemented in simple Python code. You should
    generate a treatment assignment function T(X) that maps an individual with
    covariates X to the observed treatment T(X). This should be a suitably complex
    and realistic function which can be implemented in simple Python code. This DGP
    should be generated in Python code. You should then save the data into a csv
    file as follows, where columns are X_0, ..., X_n, t, y, t_test, cepo_test. n
    refers to the number of covariates we originally had. Each row represents one
    individual in the original lenta dataset

Rows X_0, X_n should retain the base covariate values from the original dataset

t = T(X) for corresponding X value (where X represents the set of covariates)

y = f(X, t) for X the corresponding covariate value and t + Gaussian noise which is
    iid for each row.

t_test should be randomly sampled from a suitable range based on the nature of the
    continuous treatment you propose

cepo_test = f(X, t_test) for the corresponding t_test This should be a python script,
    with a main() function. You should also randomly sample only 4000 rows to
    include in the csv.
```

## E. Ablation Studies

We perform ablations on important design choices related to our prior. We find a significant performance gain from including in-pass tabular "corruption" (Sui et al., 2024; Ma et al., 2025a) (e.g. binarizing, quantizing, or zero-inflating randomly-selected nodes) as opposed to solely post-hoc (Table 3).

We also experiment with a simplified prior design that uses a single MLP (Table 4), but find that the 3-MLP design acheives greater performance, especially on the DPE metric.

Next we reconsider the necessity of imposing positivity in the prior for DGP generation. Positivity is one of the assumptions needed for identifiability of the treatment-response curve in theory. Indeed, we find that positivity is a necessary factor, and the prior without it generates data that is less effective for training CCPFN (Table 5).

Finally, instead of the histogram variant of the causal data-prior loss, we test a loss based on the continuous ranked probability score. In Table 6 we find that the CRPS loss gives slight improvement on TCGA, but otherwise is not as effective as the causal data-prior loss.

*Table 3.* Ablation on applying tabular "corruption" during the data-generation process v.s. only afterwards (post-hoc). Reported metrics are MISE and DPE.

| Method | Debt $(\times 10^{-2})$ | MVICU $(\times 10^3)$ | Warfarin | TCGA | News | NewsHet $(\times 10^{-2})$ |
|---|---|---|---|---|---|---|
| | Mean MISE (↓ better) | | | | | |
| **Ours** | **2.22** | **1.45** | **36.6** | **8.63** | 1.57 | **5.58** |
| Post-Hoc Corruption Only | 3.75 | 1.53 | 54 | 9.11 | **1.53** | 6.24 |

| Method | Debt $(\times 10^{-3})$ | Warfarin | TCGA | News | NewsHet $(\times 10^{-3})$ |
|---|---|---|---|---|---|
| | Mean DPE (↓ better) | | | | |
| **Ours** | **.110** | **2.62** | 38.9 | 3.80 | **1.52** |
| Post-Hoc Corruption Only | 3.11 | 2.66 | **36.5** | **3.22** | 2.52 |

*Table 4.* Ablation on the prior design. Ours is the 3-MLP prior discussed in Section 2; 1-MLP is a single MLP prior. Reported metrics are MISE and DPE.

| Method | Debt $(\times 10^{-2})$ | MVICU $(\times 10^3)$ | Warfarin | TCGA | News | NewsHet $(\times 10^{-2})$ |
|---|---|---|---|---|---|---|
| | Mean MISE (↓ better) | | | | | |
| **Ours (3-MLP)** | **2.22** | **1.45** | 36.6 | 8.63 | 1.57 | **5.58** |
| 1-MLP | 5.32 | 1.67 | **27.2** | **4.23** | **1.56** | 5.75 |

| Method | Debt $(\times 10^{-3})$ | Warfarin | TCGA | News | NewsHet $(\times 10^{-3})$ |
|---|---|---|---|---|---|
| | Mean DPE (↓ better) | | | | |
| **Ours (3-MLP)** | **.110** | **2.62** | 38.9 | **3.80** | **1.52** |
| 1-MLP | 1.22 | 3.08 | **37.8** | 4.37 | 2.50 |

*Table 5.* Ablation on enforcing positivity in the prior on DGPs. No positivity enforced entails having $\mathbf{MLP}_T$ output the observed $T$ directly, with no noise node $\eta_T$ added. Reported metrics are MISE and DPE.

| Method | Debt $(\times 10^{-2})$ | MVICU $(\times 10^3)$ | Warfarin | TCGA | News | NewsHet $(\times 10^{-2})$ |
|---|---|---|---|---|---|---|
| | Mean MISE (↓ better) | | | | | |
| **Ours** | **2.22** | **1.45** | **36.6** | 8.63 | **1.57** | **5.58** |
| No Positivity Enforced | 9.98 | 1.48 | 39.7 | **7.11** | 1.68 | 6.97 |

| Method | Debt $(\times 10^{-3})$ | Warfarin | TCGA | News | NewsHet $(\times 10^{-3})$ |
|---|---|---|---|---|---|
| | Mean DPE (↓ better) | | | | |
| **Ours** | **.110** | **2.62** | 38.9 | **3.80** | **1.52** |
| No Positivity Enforced | 886 | 3.34 | **36.0** | 4.07 | 3.16 |

*Table 6.* Ablation on choice of loss function (CRPS v.s. CE). Reported metrics are MISE and DPE.

| **Method** | Debt $(\times 10^{-2})$ | MVICU $(\times 10^{3})$ | Warfarin | TCGA | News | NewsHet $(\times 10^{-2})$ |
|---|---|---|---|---|---|---|
| | **Mean MISE ($\downarrow$ better)** | | | | | |
| **Ours (CE)** | **2.22** | **1.45** | **36.6** | 8.63 | **1.57** | **5.58** |
| CRPS | 3.06 | 1.57 | 38.9 | **8.03** | 1.58 | 5.90 |

| **Method** | Debt $(\times 10^{-3})$ | Warfarin | TCGA | News | NewsHet $(\times 10^{-3})$ |
|---|---|---|---|---|---|
| | **Mean DPE ($\downarrow$ better)** | | | | |
| **Ours (CE)** | **.110** | **2.62** | 38.9 | **3.80** | **1.52** |
| CRPS | .389 | 2.73 | **37.2** | **3.80** | 4.5 |

