# OpenReview forum: "Causal Foundation Models with Continuous Treatments"
_ICML.cc/2026/Workshop/FMSD — FMSD @ ICML 2026 Poster_

### Official Review · Reviewer_Pc33 · 2026-05-18
**Well-written paper proposing and providing some initial experimental results for training a causal foundation model targeting the continuous treatment setting.**

**Rating:** 7
**Confidence:** 3

**Review:**

### Summary

The paper proposes a causal foundation model for the continuous treatment setting. By training the model on synthetic data from different data generating processes with continuous treatments, the authors hope to obtain a strong prior for prediction of treatment effects from observational data. The authors present some preliminary results demonstrating that this idea is promising and provides performance improvements in some scenarios.

### Strengths

- The paper is clear and well-written, explaining the scope and objectives well.
- The paper addresses an important and difficult problem (treatment effect estimation in the continuous treatment setting).
- In order to address this problem, the authors design a synthetic data generating process which respects the assumption and principles of causal inference (the 3-MLP prior).
- The authors evaluate their proposed method across different datasets and compare to different baselines, using two different metrics.

### Areas for Improvement

- **3-MLP and imposed priors:** While 3-MLP seems like a principled way for generating synthetic training data for causal inference, it is unclear to me what exact priors the authors aim to impose on the model with this setup. The type of functions (in terms of continuity, smoothness, or monotonicity) that can be randomly generated with 3-MLP seems quite constrained and not precisely characterised. Hence, I am wondering whether CCPFN trained with this method is more suitable for certain datasets (created in similar ways) than for others. In other words, I think it would be great to discuss in what circumstances the prior learned by CCPFN might be helpful to the downstream tasks, and in what circumstances it would actually be counterproductive. Are there any settings when using CCPFN would lead to worse performance than not using any prior? Characterising such settings is important for safe deployment of the proposed method.
- **Experiments:** The experimental results provided are interesting, and show the method to be promising, but seem quite preliminary. In particular, it seems that the proposed method (CCPFN) provides performance improvements in some but not all settings. The performance improvements are also visible when evaluating MISE, but not DPE. Given the large variability in the results, it would be great if the authors could explain (at least partially) in what settings is CCPFN particularly helpful, and when the downstream users should in turn resort to other methods because CCPFN might lead to bias.
- **Appendix:** I would encourage the authors to make the appendix more self-contained, by including more detailed description of the architecture used and the tokenisation scheme.

### Detailed Comments

I would encourage the authors to run an ablation showing how the performance of CCPFN improves compared to other methods, as the size of the observational dataset $\mathcal{D}_{\mathrm{obs}}$ increases. The intuition here is that CCPFN, if it learns correct priors, should be particularly effective in the small data regime.

### Justification of Score

I think this is a well-written paper which addresses an important problem. However, the experimental results seem not complete yet, and I would encourage the authors to further explore when the proposed method should provide particular performance benefits, and when it can be hurtful to the downstream performance (for which more case studies and ablations would be needed). In the current form, the claims about “state-of-the-art” performance in the abstract seem a bit of an overstatement.

---

### Official Review · Reviewer_Ne3x · 2026-05-19
**Continuous Causal Prior-Fitted Network**

**Rating:** 7
**Confidence:** 4

**Review:**

## Summary

This paper introduces CCPFN (Continuous Causal Prior-Fitted Network), the first causal foundation model designed specifically for the continuous treatment setting. The authors address the gap that existing ICL-based causal foundation models (CausalPFN, Do-PFN, CausalFM) focus on binary treatments, while many real-world applications require estimating individual treatment-response curves (ITRCs) across continuous treatment values.

## Strengths

- **Novelty and timeliness**: To my knowledge, this is genuinely the first causal foundation model for continuous treatments, filling an important gap. The transition from binary to continuous treatments is non-trivial and the contribution is well-motivated.
- **Principled prior design**: The 3-MLP construction explicitly encodes identifiability assumptions (unconfoundedness via backdoor structure; positivity via heteroscedastic noise on $T$). The ablation in Table 5 nicely demonstrates that enforcing positivity is empirically necessary, not just theoretically motivated.
- **Architectural innovation**: The separate nonlinear T-encoder is a sensible response to the dimensionality concern that treatment signal can get drowned out in high-dimensional covariate spaces. The S-Learner-inspired design (appending $t$ to $\mathbf{x}$) is well-justified.
- **Strong empirical results**: CCPFN achieves top average rank on MISE (2.8 vs. 3.3 for the next-best method), and notably without per-dataset tuning, which is a meaningful practical advantage.


## Areas for Improvement

- **DPE results are underwhelming**: While MISE results are strong, CCPFN ranks 6th on DPE (Table 2), behind ADMIT, TabDPT, TabPFN, and TabICL. The abstract claims "state-of-the-art performance" but this is only true for MISE. The paper should more honestly characterize this trade-off and discuss why a model with the lowest MISE doesn't translate to better dosage policy error as this is an important practical limitation.
- **Limited theoretical analysis**: The paper claims the 3-MLP prior "directly encodes identifiability" but the argument is informal. A more rigorous statement of which class of SCMs the prior covers, and whether the model can in principle recover ITRCs for SCMs outside this class, would strengthen the contribution.
- **No real-data evaluation**: All benchmarks are semi-synthetic or fully synthetic. Given that this is a foundation model meant for "off-the-shelf" deployment, even one real-world case study where ground truth or expert assessment is available would substantially strengthen the contribution.
- **Variance is high**: Standard deviations on MVICU and Warfarin are large relative to differences between methods (e.g., MVICU: 1.45 ± 2.4). The narrative of "first place" rests on differences that are within noise for several datasets. Statistical significance testing or rank-based confidence intervals would help.
- **Treatment range assumption**: The model maps treatments to $[0, 1]$ via min-max scaling, which assumes treatments observed at training/test time span the relevant range. This may break under covariate shift in treatment distribution.
- **Comparison to Bernstein/Value-based priors**: Appendix C.2 mentions alternative priors but no quantitative comparison is provided. Why are these described if not compared?


## Detailed Comments

1. The "tabular corruption during forward pass" innovation is interesting and the intuition that downstream SCM mechanisms can adapt to realistic discreteness is compelling. However, the 35% during-pass / 65% post-hoc split seems arbitrary. Did you ablate this fraction?
2. The loss function uses regression-as-classification with $L=1024$ bins on $[-10, 10]$ after z-standardization. How does this discretization interact with extrapolation when test outcomes lie outside the training range?
3. For the validation dataset construction, you used LLM-generated DGPs (Appendix D.2). This is creative but raises concerns about whether validation data is genuinely independent. If the LLM has seen similar benchmark constructions, the validation set may not test true generalization. Some discussion of this risk would be appropriate.
4. The connection to CausalPFN architecture is mentioned but the actual differences (beyond the T-encoder) are not articulated clearly. A more explicit comparison table would help.
5. Did you investigate sensitivity to the number of context samples? How does performance change with fewer in-context examples?
6. How does the model behave at the boundaries of the treatment range $[0, 1]$? Figure 3 suggests good behavior near endpoints, but is this evaluated systematically?


## Justification of Score

This is a solid, well-executed paper that makes a clear contribution by being the first ICL-based causal foundation model for continuous treatments. The methodology is principled, empirical results on MISE are strong, and the prior design is genuinely thoughtful. However, the DPE results are notably weaker, the lack of real-data evaluation limits the strength of "off-the-shelf" claims, and the theoretical analysis is informal. For a workshop venue focused on foundation models for structured data, the topical fit is excellent and the contribution is novel.

---

### Official Review · Reviewer_VAGi · 2026-05-19
**Useful paper, but some claims should be scoped carefully**

**Rating:** 6
**Confidence:** 4

**Review:**

Summary:
The paper proposes CCPFN, a PFN-style causal model for continuous treatments. Instead of estimating a binary treatment effect, the model predicts individual treatment-response curves from observational data using in-context learning.

Strengths:
This is a good fit for the workshop. Continuous treatments are important, and most causal PFN work is still focused on binary treatments. I liked the 3-MLP prior because it separates covariates, treatment assignment, and outcomes in a clean backdoor setup. The paper also has useful comparisons and ablations, not just a method proposal.

Areas for Improvement:
My main concern is that the performance claim should be scoped more carefully. The MISE results look good, but the DPE results are more mixed. In Table 2, CCPFN is not the best method for dosage policy error, and ADMIT has a better average rank. So I think the results mainly support strong treatment-response curve reconstruction, not uniformly strong continuous-treatment performance.

Detailed Comments:
The benchmark setup also needs more discussion. Several datasets are semi-synthetic, and the treatment/outcome functions are generated. This is understandable for this problem, but the paper should say more about how much the results depend on those generated DGPs. A runtime or inference-cost comparison would also be useful.

Justification of Score:
I think this is a relevant workshop paper with enough empirical support. My concerns are mainly about claim calibration and benchmark realism, not the basic method. I rate it slightly above the acceptance threshold.